# Matrix condition mediates the effects of habitat fragmentation on species extinction risk

Juan Pablo Ramírez-Delgado [1✉], Moreno Di Marco[2], James E. M. Watson [3,4], Chris J. Johnson[1], Carlo Rondinini [5], Xavier Corredor Llano[1], Miguel Arias[1] & Oscar Venter[1]

Habitat loss is the leading cause of the global decline in biodiversity, but the influence of human pressure within the matrix surrounding habitat fragments remains poorly understood. Here, we measure the relationship between fragmentation (the degree of fragmentation and the degree of patch isolation), matrix condition (measured as the extent of high human footprint levels), and the change in extinction risk of 4,426 terrestrial mammals. We find that the degree of fragmentation is strongly associated with changes in extinction risk, with higher predictive importance than life-history traits and human pressure variables. Importantly, we discover that fragmentation and the matrix condition are stronger predictors of risk than habitat loss and habitat amount. Moreover, the importance of fragmentation increases with an increasing deterioration of the matrix condition. These findings suggest that restoration of the habitat matrix may be an important conservation action for mitigating the negative effects of fragmentation on biodiversity.

---

[1] Natural Resources and Environmental Studies Institute, University of Northern British Columbia, Prince George V2N 4Z9, Canada. [2] Department of Biology and Biotechnologies, Sapienza University of Rome, 00185 Rome, Italy. [3] School of Earth and Environmental Sciences, University of Queensland, St Lucia 4072, Australia. [4] Centre for Biodiversity and Conservation Science, School of Biological Sciences, The University of Queensland, Brisbane 4072 QLD, Australia. [5] Global Mammal Assessment Program, Department of Biology and Biotechnologies, Sapienza University of Rome, Rome 00185, Italy. ✉email: delgado@unbc.ca

Although habitat loss is the leading cause of the ongoing biodiversity crisis[1–4], the degree to which habitat fragmentation, defined as the spatial arrangement of remaining habitat for a given amount of habitat loss, influences the loss of biodiversity has remained the focus of considerable debate[5–10]. Central to the debate has been a persistent uncertainty in disentangling the effects of habitat loss on biodiversity from the effects of fragmentation per se, especially relative to the reduction in patch size and the increase in patch isolation[8,11]. While some studies have challenged the assumption of the impacts of fragmentation[6,8,9,12], others have demonstrated that the effects of fragmentation are negative and stronger for local species[5,7,13,14], particularly in the tropics[15] and at intermediate (30–60%) levels of habitat amount[16,17]. Resolving this debate is critical to not just informing efforts to prioritise the protection and management of intact and fragmented landscapes with the same total amount of habitat, but also to better understand the role of the areas surrounding patches of habitat, commonly referred to as 'the matrix', in maintaining biodiversity[10,18,19].

The traditional characterisation of landscapes, which view patches of habitat as islands embedded in a matrix of 'non-habitat', as assumed in classical theoretical models[20,21], has been strongly criticised[18,22–24]. This characterisation has progressively been relaxed with approaches based on the premise that the matrix should be treated as a heterogeneous mosaic of different land covers (e.g. 'countryside biogeography'[25], and the 'land-sharing' and 'land-sparing' approaches[26–30]), as it is recognised that species use different matrices for foraging, dispersing, and reproduction purposes[31,32]. While high-contrast matrices (e.g. intensive agricultural or built environments) act as movement barriers or ecological traps with an elevated risk of mortality for many species[33], low-contrast matrices (e.g. secondary forests or shade-grown low-intensive agriculture in forested regions) may act as permeable barriers with a reduced risk of mortality for many others, even for those typically considered as habitat specialists[25,27]. To date, however, conservation and management assessments have focused mainly on species' primary habitat[34,35], limiting our understanding of their response to the habitat matrix, which may have direct implications for the design of functional landscapes[36] and the prioritisation of conservation actions in fragmented landscapes[37].

Comparative extinction risk modelling is an approach for assessing the drivers of extinction risk and the change in risk over time. These models are based on the relationship between species' life histories, the habitat pressures within species geographic ranges, and their threat status[38–43]. Built with readily available data, this approach allows for the prediction of the risk of extinction of a larger number of species compared with that provided by expert-based assessments. This more rapid approach can substantially reduce resource requirements, as well as proactively inform conservation and management strategies[44,45]. However, despite the fact that the loss and fragmentation of habitat are among the main determinants of species extinction risk[3,43,46,47], the influence of the matrix condition on the effects of fragmentation and its relationship with the risk of extinction has not been well evaluated for any animal taxon at a global scale.

Here, we quantify the relationship between changes in the extinction risk of 4426 terrestrial mammals over a 24-year period (1996–2020), the fragmentation of their suitable habitat (in terms of the degree of fragmentation and the degree of patch isolation), and the levels of human pressure within the associated habitat matrix. Our goal is to test the influence of human pressure within the matrix on the effects of fragmentation for determining changes in species extinction risk globally. We focus on terrestrial mammals as they have been used as a focal taxon in previous extinction risk analyses[48], they are known to be sensitive to

fragmentation[46], and data are available to delineate levels of suitable habitat (i.e. high and medium habitat suitability) and unsuitable habitat (i.e. the matrix) within their ranges[49]. For each species, we quantify the degree of fragmentation as the average Euclidean distance into 'core' suitable habitat from the nearest patch edge, the degree of patch isolation as the average Euclidean distance between patches of suitable habitat through the surrounding matrix, and the matrix condition as the extent of high human pressure levels overlapping with areas of unsuitable habitat. Spatial data representing the condition of the matrix were obtained from the recently updated human footprint maps[50,51], which provide a single metric of the combined area and intensity of human activities, all of which are driving the current biodiversity crisis[3]. We define a human footprint threshold of ≥3 out of 50 to represent the extent of human-modified habitat within the matrix. This threshold was used as it has shown to be the strongest predictor of transitions in extinction risk for terrestrial mammals[43]. Furthermore, this human footprint threshold is associated with the highest declines in mammalian movements[52]. Following previous studies[42,43,53], we classify species into two groups of extinction risk, 'low-risk' transitions and 'high-risk' transitions, based on the first and last Red List category registered between 1996 and 2020. In combination with other predictors of extinction risk (see Table 1 for a description), we quantify the relative importance of the degree of fragmentation, the degree of patch isolation, and the condition of the matrix for determining extinction risk transitions in terrestrial mammals.

Our analyses reveal that the condition of the matrix plays a major role in the effects of fragmentation for predicting extinction risk transitions in terrestrial mammals. Our results suggest that the negative effects of fragmentation may be somewhat mitigated when the matrix is associated with lower levels of human pressure.

## Results

**Changes in species extinction risk**. We found that 2,984 (67.4%) terrestrial mammals faced a low-risk transition and 1442 (32.6%) a high-risk transition between 1996 and 2020 (Fig. 1). A total of 4124 species (93.2%) retained the same Red List category, while 302 (6.82%) changed their category through time (Supplementary Fig. 1).

**Predicting transitions in species extinction risk**. We used a Random Forest model for classification[54] to measure the performance of an array of pressure, environmental and life-history variables (see Table 1 for a description) for the prediction of extinction risk transitions in terrestrial mammals. We found that the degree of fragmentation of suitable habitat had higher predictive performance than species life-history traits, human pressure variables and other environmental conditions (Fig. 2).

Interestingly, our results show that the degree of fragmentation, the extent of high human footprint values in the matrix, and the degree of patch isolation had higher predictive performance than the change in high human footprint values (as defined by increases in high human footprint values through time) within the suitable habitat and the proportion of suitable habitat (Fig. 2). This result was supported by a sensitivity analysis where a different combination of the levels of habitat suitability was applied (Supplementary Fig. 2a, b). This suggests that habitat fragmentation and the matrix condition better predict changes in species extinction risk than habitat loss and habitat amount at a global scale.

Partial dependence plots show that the probability of high-risk transitions is higher with an increasing degree of fragmentation (Fig. 3a), a decreasing degree of patch isolation (Fig. 3b), and an extent of high human footprint values within the matrix of 100% (Fig. 3c).

**Table 1 Description of the selected variables to predict extinction risk transitions in terrestrial mammals.**

| Class | Variable | Description | Source |
|---|---|---|---|
| Pressure | High human footprint extent in the matrix | Proportion of unsuitable habitat overlapping with high human footprint values in 2000. | 49,51 |
| | High human footprint extent in patches of suitable habitat | Proportion of suitable habitat overlapping with high human footprint values in 2000. | 49,51 |
| | High human footprint change in the matrix | Difference in the proportion overlap between the area of unsuitable habitat and high human footprint values during 2000 and 2013. | 49,51 |
| | High human footprint change in patches of suitable habitat | Difference in the proportion overlap between the area of suitable habitat and high human footprint values during 2000 and 2013. | 49,51 |
| Environment | Degree of habitat fragmentation | Average of the Euclidean distance from the edge to the 'core' (i.e. the interior) of each patch of suitable habitat. | 49,82 |
| | Degree of patch isolation | Average of the Euclidean distance between patches of suitable habitat from the edge to the 'core' (i.e. the interior) of each area of unsuitable habitat. | After Crooks et al.[46] |
| | Proportion of suitable habitat | Proportion of suitable habitat within the range of each species. | 49 |
| | Realm | Biogeographic realm in which the species can be encountered. | 79,85 |
| Life-history | Body mass | A generic proxy of species life history and energetic requirements. | 100–104 |
| | Diet | Dietary categories: vertebrate carnivore (>90% vertebrate matter ingested), invertebrate carnivore (>90% invertebrate matter ingested), omnivore (10–90% animal matter ingested or 10–90% plant matter ingested), herbivore (>90% plant matter ingested). | 104–106 |
| | Weaning age | A proxy of species reproductive timing. | 101,102 |
| | Gestation length | A proxy of species reproductive output. | 101,102 |
| | Order | Species taxonomic order. | 85 |

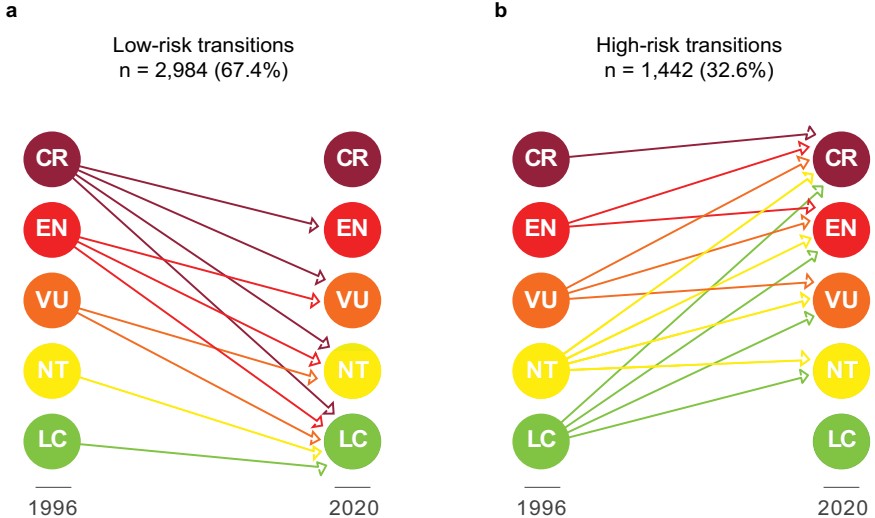

**Fig. 1 Classification of species extinction risk transitions based on past and present IUCN Red List categories[*]. a** Low-risk transitions included species that retained a category of least concern, together with those species that moved from any higher category of threat to a lower category between 1996 and 2020. **b** High-risk transitions included all species that retained a category of threatened or near threatened, together with those species that moved from any lower category of threat to a higher category between 1996 and 2020. [*]Acronyms refer to the IUCN Red List categories, including Least Concern (LC), Near Threatened (NT), Vulnerable (VU), Endangered (EN), and Critically Endangered (CR).

Our model showed good overall classification ability during cross-validation, with an accuracy of 81.2%. The proportion of correctly classified high-risk transitions (sensitivity = 60.5%) was lower than the proportion of correctly classified low-risk transitions (specificity = 90.9%), with a true skill statistic of 0.51.

The predictive performance of our model did not markedly change compared to the model built with a different combination of the levels of habitat suitability (Supplementary Table 1). Thus, the model is robust to changes in the levels of habitat suitability.

**The influence of the matrix condition on the importance of habitat fragmentation for predicting extinction risk transitions.** In order to measure the influence of the matrix condition

on the importance of fragmentation (i.e. the degree of fragmentation and the degree of patch isolation) for the prediction of extinction risk transitions, we first discretized the extent of high human footprint values within the matrix of each species into two broad levels as a proxy for matrix quality: 'low-quality' matrices and 'high-quality' matrices. As the global distribution of the matrix condition showed to be uneven in both low-risk and high-risk species (Supplementary Fig. 3), even at the scale of individual biogeographic realms (Supplementary Fig. 4), we defined cutoff values for each of the levels of quality of the matrix based on the positive and negative effect that the matrix condition had on the probability of high-risk transitions (Fig. 3c). Low-quality matrices were therefore represented by species with extents >84.2% of their

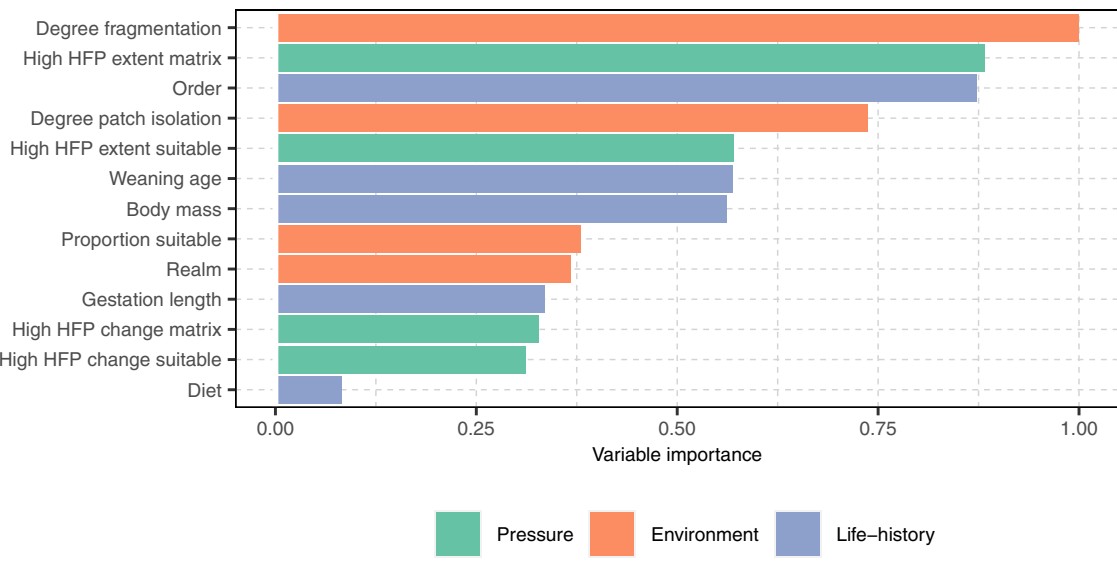

**Fig. 2 Relative importance of selected variables for the prediction of extinction risk transitions in terrestrial mammals.** Variables are colour-coded according to their broad class (human pressure, environment and life history). The description of each variable can be found in Table 1. High levels of the human footprint (HFP) included values of 3 or above. Source data are provided as a Source Data file.

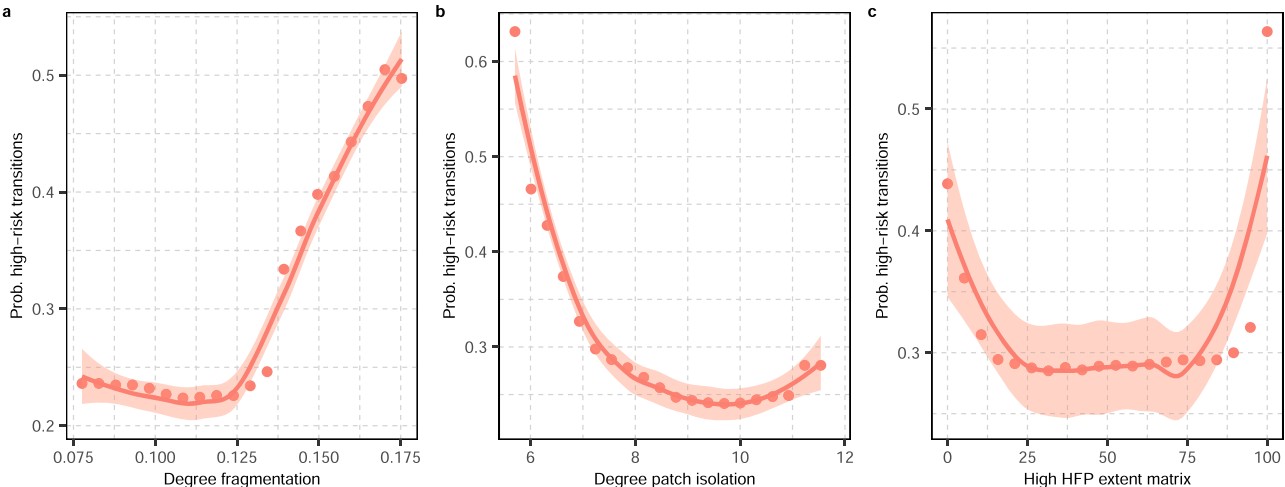

**Fig. 3 Partial dependence plots to show the effect of the degree of habitat fragmentation, the degree of patch isolation, and the matrix condition on extinction risk transitions in terrestrial mammals.** The plots show the probability of high-risk transitions as a function of **a** the degree of fragmentation, **b** the degree of patch isolation, and **c** the extent of high human footprint values within the matrix. Solid red lines and shading represent fitted LOESS curves and 95% credible intervals for the relationships between the probability of high-risk transitions and each explanatory variable. As partial dependence plots for Boolean response variables mirror each other, the probability of low-risk transitions as a function of these variables are not depicted in the figure. Values of the degree of fragmentation and the degree of patch isolation were ln-transformed for visual purposes. The degree of fragmentation was inverse-coded so high values represent high degrees of fragmentation. High values of the degree of patch isolation represent high degrees of isolation between patches of suitable habitat. The description of each variable is given in Table 1. High levels of the human footprint (HFP) included values of 3 or above. Source data are provided as a Source Data file.

matrix overlapping with high human footprint values ($n = 1815$ low-risk species and 1027 high-risk species), while high-quality matrices by those species with extents <15.8% of their matrix overlapping with high human footprint values ($n = 60$ low-risk species and 29 high-risk species). We then built separate Random Forest models for each level of quality of the matrix in order to compare the relative importance of the degree of fragmentation of suitable habitat and the degree of isolation of patches of suitable habitat between species with a matrix of low-quality habitat and species with a matrix of high-quality habitat.

We found that the degree of fragmentation and the degree of patch isolation had higher relative importance for species with a low-quality matrix (Fig. 4a) than that observed for species with a high-quality matrix (Fig. 4b), with a decrease of 33.3% and 62.5%, respectively. Notably, the relative importance of the extent of high human footprint values in the matrix was markedly higher for those species with a low-quality matrix than for those with a high-quality matrix, with a decrease of 116.4%, suggesting that the lower the quality of the matrix, the higher the predictive importance of the matrix for predicting extinction risk transition in terrestrial mammals.

When looking at the difference in the degree of fragmentation and the degree of patch isolation between low-risk and high-risk species with a low-quality matrix, we found that both variables

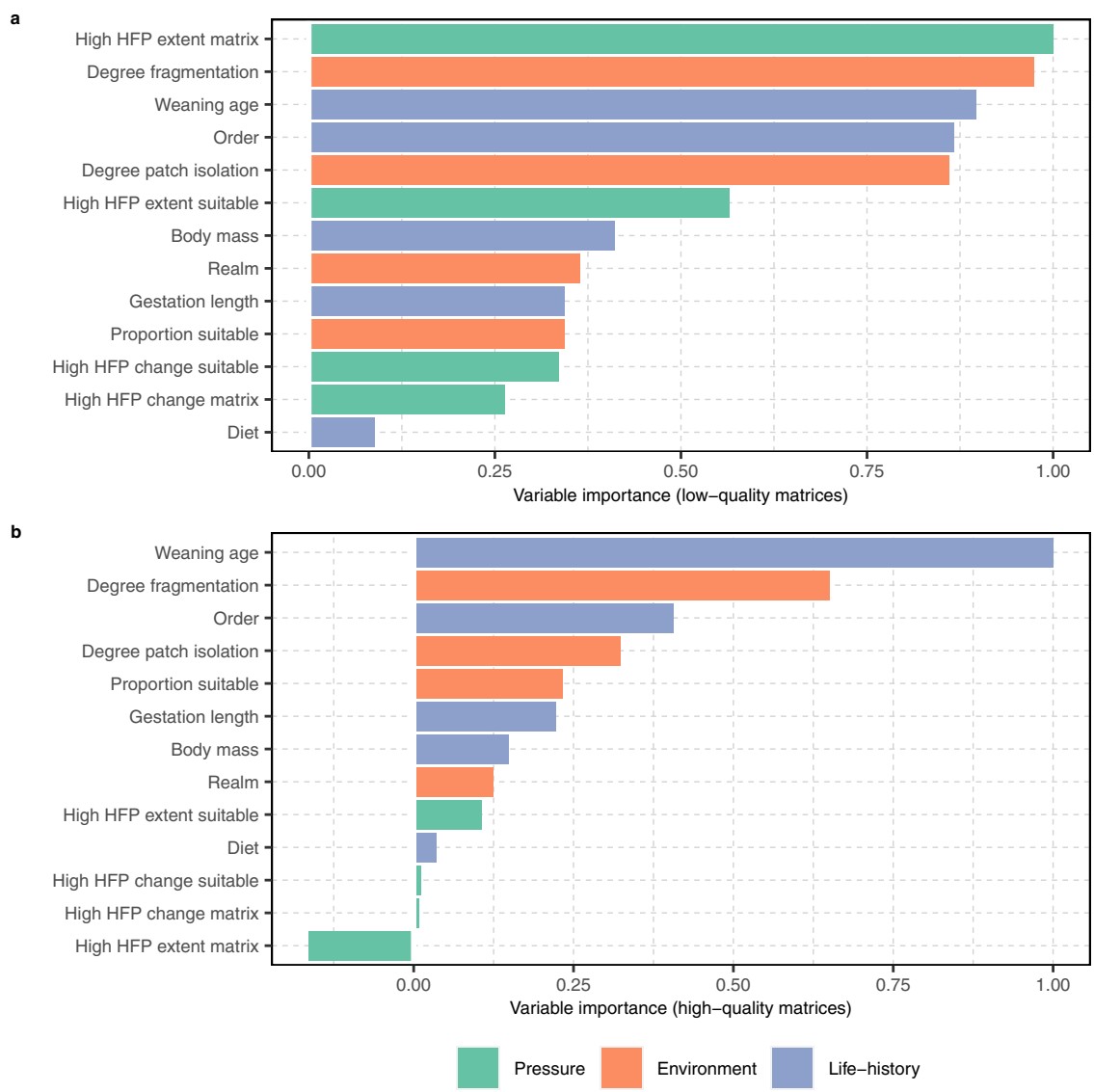

**Fig. 4 Influence of the matrix condition on the relative importance of selected variables for the prediction of extinction risk transitions in terrestrial mammals. a** Relative importance of each predictor for species with a low-quality matrix, which included proportions >84.2% of the extent of their matrix overlapping with high human footprint values ($n = 1815$ low-risk species and 1027 high-risk species). **b** Relative importance of each predictor for species with a high-quality matrix, which included proportions <15.8% of the extent of their matrix overlapping with high human footprint values ($n = 60$ low-risk species and 29 high-risk species). Variables are colour-coded according to their broad class (human pressure, environment, and life-history). The description of each variable is given in Table 1. High levels of the human footprint (HFP) included values of 3 or above. Source data are provided as a Source Data file.

were significantly higher for those species classified as high-risk ($p$-values < 0.001; Wilcoxon rank-sum test, one-sided). We also found that the difference of the degree of fragmentation and the degree of patch isolation between low-risk and high-risk species with a high-quality matrix was not statistically significant ($p$-values > 0.05; Wilcoxon rank-sum test, one-sided). The degree of fragmentation and the degree of patch isolation showed a greater effect size between low-risk and high-risk species with a matrix of low-quality habitat, with an estimated Cohen's d of 0.23 and 0.32, respectively (Supplementary Fig. 5a, b). This indicates a greater effect of both the degree of fragmentation and the degree of isolation between patches of suitable habitat on the risk of extinction of those terrestrial mammals with a matrix of low-quality habitat.

Our results show that the model for species with a matrix of low-quality habitat had higher predictive performance (true skill statistic = 0.57) than the model for species with a matrix of high-quality habitat (true skill statistic = 0.36). Our results also show that the classification ability was higher in the model for species with a low-quality matrix (accuracy = 0.81) compared to that shown in the model for species with a high-quality matrix (accuracy = 0.76). Although the proportion of correctly classified high-risk transitions was higher in the model for species with a low-quality matrix (sensitivity = 68.4%), relative to that shown in the model for species with a high-quality matrix (sensitivity = 42.9%), the model for species with a low-quality matrix had a lower proportion of correctly classified low-risk transitions (specificity = 88.7%) than the model for species with a high-quality matrix (specificity = 92.9%). This indicates a higher imbalance between the proportion of low-risk species and high-risk species correctly classified in the model for species with a high-quality matrix.

## Discussion

Understanding the external conditions under which a species is likely to face an increased risk of extinction are necessary to inform conservation policies and management strategies[42]. We found that the condition of the matrix, as defined by the extent of high human footprint values between patches of suitable habitat, strongly influenced the effects of fragmentation on extinction risk transitions of terrestrial mammals. Specifically, we found that the degree of fragmentation and the degree of patch isolation had a higher relative importance for species with a matrix of low-quality habitat compared to those with a matrix of high-quality habitat when determining extinction risk transitions in terrestrial mammals. To the best of our knowledge, these findings are the first to demonstrate the extent to which human pressure within the matrix alters the importance of fragmentation metrics as predictors of extinction risk transitions in terrestrial mammals. These findings are in line with previous studies showing that the use of the matrix is among the main determinants of the vulnerability of mammalian populations to local extinction in fragmented landscapes (e.g. refs. [25,55,56]), and supports recent findings showing that species-area relationships are steeper (i.e. more extinction driven) in forested landscapes with a low-quality matrix, and shallower (i.e. less extinction driven) in those forested landscapes with a higher quality matrix[57]. This suggests that the magnitude of the effects of fragmentation depends on the structural similarity between suitable habitat patches and the matrix, as also suggested by a growing body of evidence across multiple taxa on a local scale[31,58].

Our results showed that species with a greater degree of fragmentation, lower degree of patch isolation, and lower quality matrix within their ranges tended to be at greater risk of extinction. This indicates that the persistence of terrestrial mammals depends not only on the proportion of suitable habitat and its spatial configuration, but also on the quality of the matrix. This might be related to the fact that species occurring in regions with low rates of historical disturbance are more likely to be sensitive to fragmentation[15], and thus more likely to face an increased risk of extinction. That would suggest that those species within the high-risk group are mainly concentrated in the tropics, particularly in forested landscapes where deforestation continues at a rapid rate[59,60]. In our sample, the majority of these species (68.4%) were restricted to the Neotropical (32.0%), Afrotropical (34.9%) and Indo-Malay (48.0%) biogeographic realms, which is consistent with these findings.

In our extinction risk model, some variables had higher predictive performance than others. For example, the degree of fragmentation of suitable habitat showed to be the most important predictor of changes in species extinction risk when compared with species life-history traits, measures of human pressure, and other environmental conditions. This finding is in line with previous extinction risk modelling showing that the inclusion of the degree of fragmentation as a predictor increases the explanatory power of the models[46,47]. In particular, this result supports recent findings showing that terrestrial mammals with higher degree of fragmentation have smaller ranges, lower proportions of suitable habitat, and are at greater risk of extinction[46].

The second most important predictor of extinction risk transitions was the extent of high human footprint values within the matrix (i.e. the condition of the matrix). This result contrasts with the findings from previous extinction risk modelling exercises for mammals, where the predictive importance of human pressure was found to be lower than life-history traits or environmental conditions different to fragmentation[48,61,62]. However, it complements the findings of one recent extinction risk modelling exercise for mammals[43], where the extent of high human footprint values within species' ranges had higher predictive importance than species life-history traits, environmental conditions (without consideration of habitat fragmentation), and other pressure variables. Given that species are not homogeneously distributed throughout their ranges[49,63], this result specifically suggests that the condition of the matrix surrounding patches of suitable habitat is strongly correlated with extinction risk transitions in terrestrial mammals. This may in part be explained by the fact that habitat loss and fragmentation have opened up the path to a series of other threat mechanisms through the matrix, such as hunting, disease spread, and invasive species[4,5,40]. It may also be related to the fact that species are increasingly obligated to inhabit human-modified landscapes[64,65], many of which have a matrix that likely prevents their movement[52] and elevates their mortality[33] (e.g. by roadkill[66] or increasing predation[67,68]).

Conflicting results on the effects of fragmentation on biodiversity have arisen from studies attempting to separate 'independent' effects of habitat loss from those of habitat fragmentation[11]. Some studies have argued that the effects of habitat loss are greater and more negative (e.g.[6,8,9,12]), while others have demonstrated that the effects due to fragmentation (such as declining patch size, increasing habitat isolation, and increasing edge effects) are essentially negative and lasting (e.g.[5,7,13,14]). However, in real landscapes, habitat loss inevitably causes habitat fragmentation, and both act in synergy with other threats to biodiversity[69–71]. Thus, there is little practical value in attempting to separate the effects of habitat loss and fragmentation[7,72–74]. Our study does not attempt to resolve the current debate as to whether and how habitat fragmentation per se (i.e. the spatial arrangement of remaining habitat for a given amount of habitat loss) influences biodiversity[5–10], but our results showed that the degree of fragmentation of suitable habitat, the extent of high human footprint values within the matrix, and the degree of isolation between patches of suitable habitat were more important predictors of extinction risk transitions than the change in high human footprint values (as represented by increases in high human pressure levels over time) within patches of suitable habitat and the proportion of suitable habitat. This suggests that changes in species extinction risk are primarily determined by the fragmentation of habitat and the matrix condition, and secondarily by the loss and the amount of habitat within species' ranges. However, there is also the possibility that the loss of most suitable habitat patches had already occurred before the beginning of the study period, resulting in the degree of fragmentation, the matrix condition, and the degree of patch isolation being more important predictors of extinction risk than habitat loss and habitat amount.

Although biogeography was not a key parameter for determining extinction risk transitions in our models, we found some differences in the way the matrix condition was distributed in low-risk and high-risk species among biogeographic realms. The Indo-Malay realm represented a particular case, with a highly right-skewed distribution (i.e. towards a higher extend of high human footprint values within the matrix) in species classified as low-risk, very similar to that shown in high-risk species. With ~87% of terrestrial mammals showing a low-quality matrix in the Indo-Malayan realm, this might indicate that species living in the Indo-Malayan realm are relatively more resilient to those human activities included in the human footprint, but more vulnerable to other threats (such as overexploitation, relevant in Southeast Asia[75]), as also suggested by others[43]. The Australasia realm also represented a particular case, with an approximately bimodal distribution of the matrix condition in low-risk species and a less right-skewed distribution in those classified as high-risk relative to that shown in other realms. Interestingly, the Australasia realm showed a lower proportion of species with a low-quality matrix (37.7%) when compared to other realms, suggesting that species

restricted to this realm have relatively lower levels of human activity within their matrix. This is perhaps unsurprising given the fact that most recent declines of Australian terrestrial mammals have occurred in areas with low human population pressures, where native vegetation has not been significantly removed, particularly in the interior deserts and tropical savannas[67]. In fact, the decline of most Australian species has been directly related to predation by introduced species (such as the feral cat, *Felis catus*, and the red fox, *Vulpes vulpes*) and changes in fire regimes[67,68], which are not included in the human footprint.

Our models were better at correctly classifying low-risk transitions than high-risk transitions. This suggests that the external conditions leading to a high-risk transition might be more difficult to identify than those leading to a low-risk transition, as also indicated in previous studies[41–43,62,76]. However, it is important to acknowledge that the exclusion of other variables associated with pressure (such as overhunting, disease, invasive species and climate change) and life-history traits (such as rarity, dispersal mode and ranging behaviour) could have increased the uncertainty of our predictions, and thus influenced the ability of our models to correctly classify high-risk transitions.

An important next step will be to create a global map by weighting the extent of the matrix of the world's terrestrial mammals with the human footprint in order to highlight those matrices with high number of species and low human pressure levels, and those with low number of species and high human pressure levels. If species threat statuses are considered, such an analysis could have the potential to identify where conservation actions are needed to be improved. For example, in those locations where species with an increased risk of extinction show a low-quality habitat within their matrix, a land-sparing approach could be effective as it maximises conservation actions on the remaining patches of suitable habitat while concentrating agriculture production elsewhere[26,28–30]. Alternatively, in those locations where species with an increased risk of extinction show a high-quality habitat within their matrix, a land-sharing approach would work better as it minimises the impact of agriculture production by maintaining or restoring the conservation value of the land already farmed[27–30].

Our results indicate that species suffering from greater pressure in their matrix require particular conservation attention. Among these species, those with smaller ranges require careful management of the areas surrounding their suitable habitat, especially in light of the current and future effects of climate and land-use change[77]. Our results also highlight the potential of high-quality matrices to mitigate the negative effects of fragmentation on species extinction risk, thus suggesting that in addition to efforts to maintain remaining suitable habitat[32,35,78], there is a need for restoration of habitats in the matrix.

## Methods

**Habitat suitability models.** We used habitat suitability models developed by Rondinini et al.[49] to represent the extent of suitable habitat patches and the extent of the matrix of 4426 out of 5709 extant terrestrial mammals, corresponding to ~78% of all species in the group[79]. The models were limited to occur within the known geographic range of each species (i.e. the current "limits of distribution of a species, accounting for all known, inferred or projected sites of occurrence", as defined by the IUCN Red List of Threatened Species[80]), and built for the year 2000 at a spatial resolution of 300 m based on species' elevation range and other habitat affinities, including preferred land cover types, tolerance to human impact, and relationship to water bodies. Species' elevation range was incorporated into the habitat suitability models when known and recorded in the IUCN Red List. Textual descriptions of other habitat affinities for each species, derived from the input of thousands of mammal experts belonging to more than 30 specialist groups of the IUCN Species Survival Commission (IUCN/SSC)[59], were also extracted from the IUCN database and input as quantitative data into the habitat suitability models. The models ranked areas with three levels of habitat suitability: (i) high, representing primary habitat or preferred habitat where the species can persist; (ii) medium, representing secondary habitat where the species can occur but not persist without nearby high suitable habitat; and (iii) unsuitable, representing locations where the species is expected to occasionally or never be found. A subset of the models and their associated levels of habitat suitability were validated against available points of known species occurrences. Full details on the development of the models are available elsewhere[49].

When delineating the levels of habitat suitability for each species, small contiguous groups of pixels (<4 adjacent pixels of the same level of habitat suitability) were removed and replaced with the pixel value of the largest and nearest group of pixels, based on eight neighbouring pixels of the same class. This reduced the influence of isolated groups of pixels of the same level of habitat suitability, and improved the computational efficiency of the analysis, as also reported in other studies[46,81].

For our analysis, we combined high and medium habitat suitability to represent the extent of suitable habitat patches, and use the level of unsuitable habitat to represent the extent of the matrix of each species. We also applied a different combination of the levels of habitat suitability when representing the extent of suitable habitat patches (high suitability instead of high and medium suitability combined) and the extent of the matrix (medium suitability and unsuitable combined instead of unsuitable habitat alone) of each species as a sensitivity analysis (see Sensitivity analysis section).

**The degree of habitat fragmentation and the degree of patch isolation as predictors of extinction risk transitions.** For each species, we measured the degree of habitat fragmentation by calculating the average Euclidean distance of all the pixels of suitable habitat from the nearest edge[82], edges demarcated by the boundary between suitable and unsuitable habitat. Large values of the average Euclidean distance represented low degrees of habitat fragmentation, whereas small values represented high degrees of habitat fragmentation. Additionally, we calculated the average Euclidean distance between patches of suitable habitat through the surrounding matrix (i.e., the average Euclidean distance of all the pixels of unsuitable habitat from the nearest edge) to account for patch isolation (after Crooks et al.[46]). Here, large values of the average Euclidean distance represented high degrees of patch isolation, and small values represented low degrees of patch isolation. The average Euclidean distance was considered because this metric does not require a predetermined distance threshold of what constitutes an edge, accounts for different shapes of fragments and landscapes patterns and arrangements, accounts for the distribution of habitat area[83], is comparable across landscapes of different extents, and provides stable and readily interpretable information[81]. Moreover, average Euclidean distance has been shown to be singularly valuable in quantifying the relationship between habitat fragmentation and extinction risk of the world's terrestrial mammals[46], which made it highly suitable for our analyses.

**The matrix condition as a predictor of extinction risk transitions.** Spatially explicit data on the condition of the matrix, as represented by the extent and change over time of high human pressure levels overlapping with the area of unsuitable habitat surrounding patches of suitable habitat (after Di Marco et al.[43]), was obtained from the recently updated global human footprint maps[51]. These maps represent the most comprehensive global distribution of changing human pressure on the environment at 1 km resolution between 2000 and 2013, based on eight pressure layers[50]: (i) built environments; (ii) intensive agriculture; (iii) pasture land; (iv) human population density; (v) night-time lights; (vi) roads; (vii) railways; and (viii) navigable waterways, all of which are driving the current extinction crisis[3]. Each human footprint map provides a single pressure metric ranging from 0 to 50, where a value of 0 represents areas free of any human influence (e.g., terrestrial remaining wilderness), values of 4 or below represent areas of low human pressure (e.g., pasture lands) and values above 20 represent areas with very high pressure levels (e.g., densely populated semi-urban and urban environments).

In this analysis, we measured the extent of high human footprint values and the change of this extent over time (between 2000 and 2013) within areas of unsuitable habitat, using a defined human footprint threshold of 3 or above. This threshold was used as it has shown to be the strongest predictor of extinction risk transitions in terrestrial mammals[43]. Moreover, this human footprint threshold is associated with the highest declines in mammalian movements[52]. Based on previous studies[43,84], we used the extent of high human footprint values within the matrix as the extent of high pressure levels within species' ranges has shown to be more sensitive to predict extinction risk than using mean values of human pressure within species' ranges. We also considered the change in the extent of high human footprint values after discarding areas where the human footprint was lower in 2013 than in 2000 (assuming no change in these particular areas), as decreases in human pressure levels are likely to take time before having a measurable effect on species threat status, particularly for species with a long generation time period[43].

**Changes in species extinction risk.** We used the IUCN Red List of Threatened Species[79,85], the retrospective Red List assessments published in Hoffmann et al.[53], and the IUCN list of genuine changes in the conservation status of mammal species (https://www.iucnredlist.org/resources/summary-statistics) to represent trends in extinction risk of terrestrial mammals. Following the classification of extinction risk transitions developed by Di Marco et al.[42,43], we classified the species into two

main groups, 'low-risk' transitions and 'high-risk' transitions (Fig. 1). The low-risk group included species that retain a category of least concern, together with those species that move from any higher category of threat to a lower category assessment period. The high-risk group included all species that retain a category of threatened or near threatened, together with those species that move from any lower category of threat to a higher category over time.

For our analysis, we classified species to the two extinction risk groups (low-risk transitions and high-risk transitions) based on the first and last Red List category registered between 1996 and 2020. In order to test the sensitivity of this classification, we also classified all species into the two extinction risk groups based on the last two Red List assessments registered between 1996 and 2020 (i.e. the second to last and last Red List categories registered during this time period). With this classification, however, only two species (0.05% of 4426 species in our sample) changed their extinction risk transition (from a high-risk transition to a low-risk transition) compared to that based on the first and last Red List category registered during the study period. We thus only reported the main results using the first and last Red List category registered between 1996 and 2020.

We excluded species without a defined level of habitat suitability, those not evaluated in the Red List, and those categorised as Data Deficient, Extinct and Extinct in the Wild in the last Red List assessment reported during the study period, as long as they have not shown a defined transition of extinction risk (see Fig. 1) along the study period.

**Predicting extinction risk transitions**. We used a multivariate Random Forest model to predict extinction risk transitions in terrestrial mammals (Fig. 1). Random Forest is a non-parametric, tree based, machine-learning technique that produces multiple decision tress using a randomly selected subset of training samples and variables to make a prediction[54,86]. Due to its limited assumptions on data distributions, its high classification stability and performance, and its ability to cope well with a large number of potentially correlated predictors and non-linear responses, Random Forest is a highly suitable technique for species threat status classification[62]. Furthermore, Random Forest modelling has shown to have the highest performance among several machine learning techniques tested for the prediction of global extinction risk of terrestrial mammals[76], which made it ideal for this study.

In this analysis, we optimised the number of trees to grow and the number of predictors sampled for splitting at each node from three repeats of 10-fold cross-validation, using 75% of the data as training data and 25% as test data. Predictors included: (i) the extent of high human footprint values in the matrix; (ii) the extent of high human footprint values in patches of suitable habitat; (iii) the change over time of high human footprint values in the matrix; (iv) the change over time of high human footprint values in patches of suitable habitat; (v) the degree of fragmentation of suitable habitat; (vi) the degree of isolation between patches of suitable habitat; (vii) the proportion of suitable habitat; and (viii) the biogeographic realm in which the species can be encountered (see Table 1 for a description). Because mammals of greater body size usually move farther[87], and diet may influence their movements as a result of differences in availability of resource types and foraging cost[88,89], we decided to include body size and dietary breadth as life-history predictors. We also included the reproductive traits weaning age and the gestation length. Other life-history traits were broadly captured by including taxonomic orders. Because the levels of habitat suitability are limited by the size of species' geographic ranges, we did not include species' range size as a predictor in order to avoid potential circularity in the estimation of extinction risk[38].

We measured the predictive importance of each variable using the mean decrease in classification accuracy (MDA) metric[54], which reports the model's ability to correctly classify data if the values of a predictor variable are randomly permuted. Based on this metric, we then calculated the relative importance of each variable using the model improvement ratio (MIR) metric[90], which scales raw importance scores from 0 to 1. Unlike the raw importance scores, the MIR metric is not influenced by the total number of variables, and is comparable among models. MIR is calculated as $[I_n/I_{max}]$, where $I_n$ is the importance of a given variable, and $I_{max}$ is the maximum model improvement score. We also reported the overall performance of the Random Forest model through cross-validation in terms of proportion of correctly classified species (accuracy), proportion of correctly classified high-risk species (sensitivity), proportion of correctly classified low-risk species (specificity) and the true skill statistic (TSS = sensitivity + specificity − 1)[91].

**Assessing the influence of the matrix on the importance of fragmentation for predicting extinction risk transitions**. To measure the influence of the matrix condition on the importance of habitat fragmentation (in terms of the degree of fragmentation and the degree of patch isolation) for the prediction of extinction risk transitions, we first defined two broad levels of quality of the matrix, 'low-quality' matrices and 'high-quality' matrices, based on the proportion of high human footprint values within the matrix of each species. When delimiting the two levels of quality of the matrix, the extent of high human footprint values in the matrix of each species was discretized into two intervals based on the positive and negative effect that the matrix condition had on the probability of high-risk transitions (see Fig. 3c). We then built separate Random Forest models for species restricted to such levels of quality of the matrix. Using the MDA metric[54] and the MIR metric[90], we measured the relative importance of the degree of habitat

fragmentation and the degree of patch isolation, including the other selected predictors of extinction risk (Table 1), from the built Random Forest models. We used cross-validated measures of accuracy, sensitivity, specificity and the true skill statistic to evaluate the overall performance of the models[91]. We also used Wilcoxon rank-sum tests to test for statistical differences in the degree of fragmentation and patch isolation between low-risk and high-risk species restricted to the defined levels of quality of the matrix. In order to determine the effect size of the degree of habitat fragmentation and patch isolation between low-risk and high-risk species for each of the levels of quality of the matrix, we used Cohen's d statistic[92].

**Sensitivity analysis**. To test the sensitivity of our model, we built additional Random Forest models based on a different combination of the levels of habitat suitability to represent the extent of suitable habitat patches (high suitability instead of high and medium suitability combined) and the extent of the matrix (medium suitability and unsuitable combined instead of unsuitable habitat alone). From these models, the relative importance of each variable was quantified using the MDA metric[54] and the MIR metric[90]. The overall performance of these models was reported through cross-validation in terms of accuracy, sensitivity, specificity and the true skill statistic[91].

All spatial analyses were performed in python using the ArcPy processing module from ArcGIS Pro 2.8.2[93]. Statistical analyses were performed in R[94], using the packages 'randomforest'[95], 'caret'[96], 'iml'[97], and 'effsize'[98].

**Reporting summary**. Further information on research design is available in the Nature Research Reporting Summary linked to this article.

## Data availability
The input dataset used to run our models of extinction risk and that support the findings of this study has been deposited in GitHub (https://github.com/juanxramirez/Matrix_condition), and mirrored on Zenodo (https://doi.org/10.5281/zenodo.5803377). The Human Footprint dataset used in this study is available for download at https://doi.org/10.5061/dryad.3tx95x6d9. Habitat suitability models for the world's terrestrial mammals are available upon request from the model developers at https://globalmammal.org/habitat-suitability-models-for-terrestrial-mammals/. Data on Red List categories through time are available upon request at https://apiv3.iucnredlist.org/. In this study, these data were accessed from R[94], using the package 'rredlist'[99]. Data on genuine changes in the Red List categories are available in Hoffmann et al.[53], and at https://www.iucnredlist.org/resources/summary-statistics. The other datasets that support the findings of this study derive from published sources, cited in the Methods section and listed in Table 1. Source data are provided with this paper.

## Code availability
The code used in the analysis presented here is available in GitHub at https://github.com/juanxramirez/Matrix_condition, and mirrored on Zenodo at https://doi.org/10.5281/zenodo.5803377.

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

## Acknowledgements

We thank Michelle Venter, Kristen Kieta and Rajeev Pillay for comments on an earlier version of the manuscript.

## Author contributions

J.P.R.-D. and O.V. conceived the study; J.P.R.-D. led the analysis and interpretation of the data under the advice of O.V., M.D.M., J.E.M.W., C.J.J., C.R., X.C.L. and M.A.; J.P.R.-D. led the writing of the manuscript with input from O.V., M.D.M., J.E.M.W., C.J.J. and C.R.

## Competing interests

The authors declare no competing interests.
