## [Peer Review File · Nature Communications]

Reviewers' Comments:

Reviewer #1:

Remarks to the Author:

Review of the paper: Matrix condition mitigates the effects of habitat fragmentation on species extinction risk

In this paper the authors describe a study based on the analysis of how the fragmentation of suitable habitat caused by habitat loss, and different quantities related to the condition (based on human footprint) of the matrix of habitat surrounding remnant patches of suitable habitat, as well as some life-history traits, are related to the transitioning of 4,327 terrestrial mammal species across IUCN red list extinction vulnerability categories from 1996 to 2000. They used a random forest model with a two-class discrete variable representing the quality of the transition (high-risk vs low-risk) across IUCN extinction vulnerability categories as a responses variable and a set of descriptors quantifying different aspects of fragmentation and habitat quality, as well as life-history traits, as predictor variables. Their results suggest that the quality of the habitat in the matrix of habitat that makes up the space between the remaining suitable patches, interacts synergistically with fragmentation to influence the type of IUCN category transition experienced by these species between 1996 and 2000.

I really like the idea behind this paper. As clearly highlighted by the authors in the manuscript, it is fundamental to gain a better understanding of the role played by the quality of the matrix of habitat left behind by habitat loss to fully understand the effects on biodiversity. Even though I think that this has to one extent or another been achieved on different previous studies, this comprehensive view of the situation provides a good overview of the problem across the globe and many species. So, even if the idea is not totally new I find this to be a good fit for Nature Communications.

However, before being able to recommend acceptance, there are two major issues that I would like to see resolved before being able to make a more informed decision:

1.- As in any other article nowadays, it is of fundamental importance that the code developed, and all datasets associated with the study are provided in an online digital repository for anyone to use, and being able to replicate these results exactly. I was not able to confirm the results presented, unfortunately, and without that I can't give a complete assessment of the study. Moreover, this needs to be open to the whole scientific community. Just stating that the code is available upon request is not enough. Please note that it must be ensured that the scripts provided are fully operational from any platform so I can run it on my machine if this manuscript comes back to me.

2.- There are a few major gaps in the description of the methods that need to be resolved before this can be published. Please see my comments about the methods section below.

SPECIFIC COMMENTS

INTRODUCTION

Line 92. Seems like there is an 'and' missing after environment? Otherwise the sentence doesn't seem well constructed.

Line 98. Why is chimpanzee behavioural activity relevant here? It only gets mentioned here and also in the methods, but doesn't really add anything. It can be removed.

RESULTS

Line 135. 'classified using the second last and last Red List category instead of the initial and final Red List' - Not clear what is the difference between 'last red list category' and 'final red list category'. Also, it would be good to clarify what is the second to last category to be able to properly understand this result.

Figure 2 and 4. Relative importance must be scaled appropriately. i.e. the sum of all importance indices should be 1.

Table 2. In the description of the variables, the average of the Euclidean distance (in the different metrics) must include an explanation of what is the average of. For example, in the fragmentation metric, is it the Euclidean distance from the centroid of each patch to the edge? If so, was the average taken across patches? Then, the 'within' is unclear.

Lines 164-165. Not clear what you mean by: "This trend was reversed to an increased probability of low-risk transitions."

Figure 3. It is not clear how / why inverting the plot gives you the scenario for low-risk transition. This should be better explained. Also, I would suggest the authors to explore better ways of representing these data. The 3D plot is really challenging to understand / visualise correctly. This kind of plots are in general hard to understand if you can't rotate them.

Lines 178-179. How do the 'sensitivity' and 'specificity' values compare? It is not clear what does it mean when one is higher than the other and viceversa.

Lines 199-200. '... predictive performance for species with low-quality matrix (Fig. 4a) than that observed for species with low-quality matrix...' – One of those 'low-quality matrix' should be high-quality matrix.

Figure 4. To be able to assess the accuracy of this figure I need to have a look at the values used for the different variables associated to each species. In particular, the High HFP extent matrix scores.

Lines 218-219. This sentence is not clear. Was the Wilcoxon comparison made between high and low risk, or between high and low quality, or combinatorially between both?

Line 228. How can you know that this is an artifact of the 'imbalance' between low-risk and high-risk? There is no indication of why this should be the case. Also, please explain how was this 'imbalance' quantified.

As mentioned above, all code and data should be made available for me to run the code automatically and look at how the analyses were conducted to be able to fully evaluate the work.

DISCUSSION

Line 256. It is the first time in the manuscript that the matrix quality has been linked to plant biomass. This should be at least mentioned, ideally well explained, much earlier.

Line 276. 'helps confirms' -> 'confirms'

Lines 277-282. It would be nice to have a few lines here explaining how this work is an improvement on previous research. This paragraph gives the impression that most of it had been done before.

Line 302. Suitable habitat was more important for what?

Lines 319-321. This statement is too vague and doesn't really say anything. Either elaborate on how exactly this can inform conservation, or just remove this line.

METHOD (this is the part referred to in my main major concern comment)

Line 354. This is too vague. You need to explain fully how was the 'expert information' used and present it in an appendix and as part of the data made available.

Lines 355 – 358. Explain how the habitat suitability levels were decided and defined.

Line 362. This is not clear. What do you mean by 'largest neighbour'? Aren't all the pixels the same size?

Line 382-383. Shouldn't it be the other way around? i.e. large values of Euclidean distance representing LARGE rather than low degrees of isolation. Also, why not using the Euclidean distance between patches?

Lines 410-413. This sentence is not clear. I couldn't understand what you meant by this. Please rephrase.

Line 449. You can get rid of the first sentence and start with 'Predictors included...'

Line 456. 'avoid potential circularity' is vague. Please explain this. Why is there circularity and how is this a way of avoiding it.

Reviewer #2:

Remarks to the Author:

The importance of the quality of the matrix of land-use types surrounding patches of focal habitat is being increasingly recognised. Many studies have found that the type of matrix can interact with and influence the effects of habitat loss and fragmentation. This manuscript investigates how the condition of the matrix, with regard to human pressures, effects the extinction risk of mammal species. By conducting this analysis on the majority of terrestrial mammals this study provides a global analysis which is useful in complementing the information gained from existing more localised studies. The study provides an important contribution to the building evidence on the significance of matrix quality when analysing habitat fragmentation. There are however a few issues that need to be addressed.

It is not completely clear how the range of a species is being defined. Does this use published distribution maps or is it based on the habitat suitability models? As matrix quality is assigned based on the human footprint of unsuitable habitat within the range of a species, how the range is defined is important. The amount and proximity of the matrix habitat to suitable habitat will be influenced by how the range maps are created. For example, if the maps are created to a high resolution with habitat suitability accounted for this will be very different to polygons constructed linking occurrence records. Please can this be clarified or made more prominent if already included.

In the study the number of species with high-quality matrix and low-quality matrix were not equal but how were species with the two matrix types distributed with regard to geography and suitable habitat type (e.g. forest, grassland). If there are correlations between matrix quality and other bio-geographic variables how does this influence the interpretation of the results?

Was structural contrast between the suitable habitat and matrix considered? More contrasting suitable and unsuitable habitat types are expected to result in stronger fragmentation effects. To a certain extent the quality of the matrix from the perspective of the focal species would therefore depend on how different it is from the suitable habitat. The majority of this is likely captured by the human footprint data used but there may be differences between biomes.

Please also see the comments below:

Line 78 – what are the indications from smaller scale studies?

Line 361 – what size are the pixels?

Reviewer #3:

None

RESPONSE TO REVIEWERS

REVIEWER 1 COMMENTS

Comment:

“Review of the paper: Matrix condition mitigates the effects of habitat fragmentation on species extinction risk

In this paper the authors describe a study based on the analysis of how the fragmentation of suitable habitat caused by habitat loss, and different quantities related to the condition (based on human footprint) of the matrix of habitat surrounding remnant patches of suitable habitat, as well as some life-history traits, are related to the transitioning of 4,327 terrestrial mammal species across IUCN red list extinction vulnerability categories from 1996 to 2000. They used a random forest model with a two-class discrete variable representing the quality of the transition (high-risk vs low-risk) across IUCN extinction vulnerability categories as a responses variable and a set of descriptors quantifying different aspects of fragmentation and habitat quality, as well as life-history traits, as predictor variables. Their results suggest that the quality of the habitat in the matrix of habitat that makes up the space between the remaining suitable patches, interacts synergistically with fragmentation to influence the type of IUCN category transition experienced by these species between 1996 and 2000.

I really like the idea behind this paper. As clearly highlighted by the authors in the manuscript, it is fundamental to gain a better understanding of the role played by the quality of the matrix of habitat left behind by habitat loss to fully understand the effects on biodiversity. Even though I think that this has to one extent or another been achieved on different previous studies, this comprehensive view of the situation provides a good overview of the problem across the globe and many species. So, even if the idea is not totally new I find this to be a good fit for Nature Communications.”

Response:

We thank the reviewer for this positive comment and the careful consideration of the conceptual message of our manuscript.

Comment:

“1.- As in any other article nowadays, it is of fundamental importance that the code developed, and all datasets associated with the study are provided in an online digital repository for anyone to use, and being able to replicate these results exactly. I was not able to confirm the results presented, unfortunately, and without that I can't give a complete assessment of the study. Moreover, this needs to be open to the whole scientific community. Just stating that the code is available upon request is not enough. Please note that it must be ensured that the scripts provided are fully operational from any platform so I can run it on my machine if this manuscript comes back to me.”

Response:

We thank Reviewer 1 for this insightful comment. We have made available all the code used in our analysis, as well as the input dataset used to run our models of extinction risk. Starting from line 571, we have now amended the code availability section of our manuscript to state:

“All the code used in this work is available in GitHub and can be accessed at https://github.com/juanxramirez/Matrix_condition.”

We have also amended the data availability section of our manuscript, starting from line 557, to say: “The input dataset used to run our models of extinction risk and that support the findings of this study has been deposited in GitHub (https://github.com/juanxramirez/Matrix_condition).”

We would like to emphasize that the habitat suitability models used in our study, which played a critical role for our spatial analyses, were derived from a different study (Rondinini et al. 2011 Philos. Trans. R. Soc. B Biol. Sci.) and are available only upon request to the data developers. Due to the large file size of these models, and the high computational performance required to process them (which, depending on the computational power available, can take several days to complete), we believe it would be impractical for reviewers to replicate all of our results. However, Dr. Carlo Rondinini, who led the development of the models and is co-author of our paper, agreed to share a small subset of these models for reviewers to assess. The subset of the habitat suitability models is available for download at https://drive.google.com/drive/folders/1bevNIINXr3WAF_hv1YIfD2944E7POrk0?usp=sharing. The code used in our spatial analyses is available at https://github.com/juanxramirez/Matrix_condition.

We would also like to point out that when we were revising our code, we saw the opportunity to increase the number of species in our sample. We had originally excluded species categorized as Data Deficient, Extinct and Extinct in the Wild at the end of the study period (i.e. in the year 1996). We have now excluded species categorized as Data Deficient, Extinct and Extinct in the Wild but at the end of the study period (i.e. in the year 2020), as long as they have not shown a defined transition of extinction risk along the study period (i.e. a low-risk transition or a high-risk transition through time between 1996 and 2020). The number of species in our sample increased from 4,327 to 4,426, which slightly altered our results. Starting from line 474, we have now amended our manuscript in the methods section to state: “We excluded species without a defined level of habitat suitability, those not evaluated in the Red List, and those categorized as Data Deficient, Extinct and Extinct in the Wild in the last Red List assessment reported during the study period, as long as they have not shown a defined transition of extinction risk (see Fig. 1) along the study period.”

Based on the above criteria, we also realized that only two species (0.05% of 4,426 species) changed their extinction risk transition when comparing our classification results obtained from the first and last Red List category registered during the study period with those based on the second to last and last Red List category registered during the study period. We therefore decided to report the main results of our study using the first and last Red List category registered during the study period for classifying extinction risk transitions. We have now amended our manuscript in the methods section, starting from line 463, to say: “For our analysis, we classified species into the two extinction risk groups (low-risk transitions and high-risk transitions) based on the first and last Red List category registered between 1996 and 2020. In order to test the sensitivity of this classification, we also classified all species into the

two extinction risk groups based on the last two Red List assessments registered between 1996 and 2020 (i.e. the second to last and last Red List categories registered during the time period). With this classification, however, only two species (0.05% of 4,426 species in our sample) changed their extinction risk transition (from a high-risk transition to a low-risk transition) compared to that based on the first and last Red List category registered during the study period. We thus only reported the main results using the first and last Red List category registered between 1996 and 2020.”

Comment:

“2.- There are a few major gaps in the description of the methods that need to be resolved before this can be published. Please see my comments about the methods section below.”

SPECIFIC COMMENTS

INTRODUCTION

Line 92. Seems like there is an ‘and’ missing after environment? Otherwise the sentence doesn’t seem well constructed.”

Response:

We thank Reviewer 1 for this suggestion. We have now amended this sentence, starting from line 92, to read: “...Spatial data representing the condition of the matrix were obtained from the recently updated human footprint maps^{50,51}, which provide a single metric of the combined area and intensity of human activities, ...”

Comment:

“Line 98. Why is chimpanzee behavioural activity relevant here? It only gets mentioned here and also in the methods, but doesn’t really add anything. It can be removed.”

Response:

Agreed and we thank Reviewer 1 for pointing this out. We had originally included this sentence in the manuscript as an example of the impact of high human pressure on behavioral diversity within a species. We have now removed it from the manuscript as suggested.

Comment:

“RESULTS

Line 135. ‘classified using the second last and last Red List category instead of the initial and final Red List’ - Not clear what is the difference between ‘last red list category’ and ‘final red list category’. Also, it would be good to clarify what is the second to last category to be able to properly understand this result.”

Response:

We thank the reviewer for pointing this out. We have fixed the text in the manuscript in a more nuanced way to clarify this point. We have now amended the methods section, starting from line 463, to say: “For our analysis, we classified species into the two extinction risk groups

(low-risk transitions and high-risk transitions) based on the first and last Red List category registered between 1996 and 2020. In order to test the sensitivity of this classification, we also classified all species into the two extinction risk groups based on the last two Red List assessments registered between 1996 and 2020 (i.e. the second to last and last Red List categories registered during this time period).”

Comment:

“Figure 2 and 4. Relative importance must be scaled appropriately. i.e. the sum of all importance indices should be 1.”

Response:

We thank Reviewer 1 for this suggestion. However, for our study, relative importance scores were calculated using the model improvement ratio (MIR) metric, which scales raw importance scores from 0 to 1. This metric was used as it is not influenced by the total number of variables, and is comparable among models. We have edited our methods section, starting from line 511, to state: “..., we then calculated the relative importance of each variable using the model improvement ratio (MIR) metric⁹⁵, which scales raw importance scores from 0 to 1. Unlike the raw importance scores, the MIR metric is not influenced by the total number of variables, and is comparable among models. MIR is calculated as $[I_n/I_{max}]$, where I_n is the importance of a given variable, and I_{max} is the maximum model improvement score.”

Comment:

“Table 2. In the description of the variables, the average of the Euclidean distance (in the different metrics) must include an explanation of what is the average of. For example, in the fragmentation metric, is it the Euclidean distance from the centroid of each patch to the edge? If so, was the average taken across patches? Then, the ‘within’ is unclear.”

Response:

Agreed and we thank Reviewer 1 for pointing this out. We have now modified the text throughout the manuscript to clarify the description of these variables as suggested. We have now amended this table (now table 1) to read: “Average of the Euclidean distance from the patch edge to the ‘core’ of each patch of suitable habitat.” for the description of the degree of habitat fragmentation, and “Average of the Euclidean distance between patches of suitable habitat from the edge to the ‘core’ of each area of unsuitable habitat.” for the description of the degree of patch isolation.

Comment

“Lines 164-165. Not clear what you mean by: “This trend was reversed to an increased probability of low-risk transitions.”

Response:

We thank Reviewer 1 for pointing this out. We have removed this sentence from the revised manuscript.

Comment:

“Figure 3. It is not clear how / why inverting the plot gives you the scenario for low-risk transition. This should be better explained. Also, I would suggest the authors to explore better ways of representing these data. The 3D plot is really challenging to understand / visualise correctly. This kind of plots are in general hard to understand if you can’t rotate them.”

Response:

We thank the reviewer for this very valid point. We have now added a new figure (Fig. 3) in the manuscript to show the effect of the degree of habitat fragmentation, the degree of patch isolation, and the matrix condition on high-risk transitions. As partial dependence plots for Boolean response variables mirror each other, these plots also give the scenario for low-risk transitions. We have now added the following sentence to our manuscript, starting from line 156, to state: “As partial dependence plots for Boolean response variables mirror each other, the probability of low-risk transitions as a function of these variables are not depicted in the figure”.

Comment:

“Lines 178-179. How do the ‘sensitivity’ and ‘specificity’ values compare? It is not clear what does it mean when one is higher than the other and viceversa.”

Response:

We thank Reviewer 1 for pointing this out. We have now edited the manuscript in the results section, starting from line 165, to state: “The proportion of correctly classified high-risk transitions (sensitivity = 60.5%) was lower than the proportion of correctly classified low-risk transitions (specificity = 90.9%), with a true skill statistic of 0.51.”

Comment:

“Lines 199-200. ‘... predictive performance for species with low-quality matrix (Fig. 4a) than that observed for species with low-quality matrix...’ – One of those ‘low-quality matrix’ should be high-quality matrix.”

Response:

We thank the reviewer for this very valid point. We have now redefined cutoff values for each of the levels of quality of the matrix (low-quality matrices and high-quality matrices) to compare the predictive importance of fragmentation (in terms of the degree of fragmentation and the degree of patch isolation) for the prediction of extinction risk transitions between species with a low-quality matrix and species with a high-quality matrix. The cutoff values have now been derived from the positive and negative effect that the matrix condition had on the probability of high-risk transitions (see Fig. 3c). Starting from line 181, we have now amended our manuscript in the results section to state: “..., we defined cutoff values for each of the levels of quality of the matrix based on the positive and negative effect that the matrix condition had on the probability of high-risk transitions (Fig. 3c). Low-quality matrices were therefore represented by species with extents > 84.2% of their matrix overlapping with high human footprint values ($n = 1,815$ low-risk species and 1,027 high-risk species), while high-quality matrices by those species with extents < 15.8% of their matrix overlapping with high

human footprint values ($n = 60$ low-risk species and 29 high-risk species). We then built separate Random Forest models for each level of quality of the matrix in order to compare the relative importance of the degree of fragmentation of suitable habitat and the degree of isolation of patches of suitable habitat between species with a matrix of low-quality habitat and species with a matrix of high-quality habitat.”

Comment:

“Figure 4. To be able to assess the accuracy of this figure I need to have a look at the values used for the different variables associated to each species. In particular, the High HFP extent matrix scores.”

Response:

We thank Reviewer 1 for pointing this out. We have now provided access to the input dataset used to run our models of extinction risk, which includes the extent of high human footprint values within the matrix of each species. The dataset has been deposited in GitHub and can be accessed at https://github.com/juanxramirez/Matrix_condition.”

Comment:

“Lines 218-219. This sentence is not clear. Was the Wilcoxon comparison made between high and low risk, or between high and low quality, or combinatorially between both?”

Response:

We thank the reviewer for raising this question. The Wilcoxon comparison was made between low-risk and high-risk species with a matrix of low-quality habitat and between low-risk and high-risk species with a matrix of high-quality habitat. We have now amended the text of our manuscript at line 213 to read: “When looking at the difference in the degree of fragmentation and the degree of patch isolation between low-risk and high-risk species with a low-quality matrix, we found that both variables were significantly higher for those species classified as high-risk (p -values < 0.001 ; Wilcoxon rank sum test, one-sided). We also found that the difference of the degree of fragmentation and the degree of patch isolation between low-risk and high-risk species with a high-quality matrix was not statistically significant (p -values > 0.05 ; Wilcoxon rank sum test, one-sided).”

Comment:

“Line 228. How can you know that this is an artifact of the ‘imbalance’ between low-risk and high-risk? There is no indication of why this should be the case. Also, please explain how was this ‘imbalance’ quantified.”

Response:

We thank the reviewer for this question. The imbalance was quantified from the difference between the proportion of correctly classified low-risk species and the proportion of correctly classified high-risk species. We have now amended the text of our manuscript at line 227 to state: “Although the proportion of correctly classified high-risk transitions was higher in the model for species with a low-quality matrix (sensitivity = 68.4%), relative to that shown in the model for species with a high-quality matrix (sensitivity = 42.9%), the model for species with a

low-quality matrix had a lower proportion of correctly classified low-risk transitions (specificity = 88.7%) than the model for species with a high-quality matrix (specificity = 92.9%). This indicates a higher imbalance between the proportion of low-risk species and high-risk species correctly classified in the model for species with a high-quality matrix.”

Comment:

“As mentioned above, all code and data should be made available for me to run the code automatically and look at how the analyses were conducted to be able to fully evaluate the work.”

Response:

We thank Reviewer 1 for this insightful comment. We have made available all the code used in our analysis, as well as the input dataset used to run our models of extinction risk. We have now amended the code availability section of our manuscript at line 571 to state: “All the code used in this work is available in GitHub and can be accessed at https://github.com/juanxramirez/Matrix_condition.”

We have also amended the data availability section of our manuscript, starting from line 557, to say: “The input dataset used to run our models of extinction risk and that support the findings of this study has been deposited in GitHub (https://github.com/juanxramirez/Matrix_condition).”

A subset of the habitat suitability models can be accessed at https://drive.google.com/drive/folders/1bevNIINXr3WAF_hv1YIFD2944E7P0rk0?usp=sharing

Comment:

“DISCUSSION

Line 256. It is the first time in the manuscript that the matrix quality has been linked to plant biomass. This should be a least mentioned, ideally well explained, much earlier.”

Response:

We thank the reviewer for pointing this out. Although we agree that this is an important consideration, it was beyond the scope of our manuscript to provide insights into how the matrix quality can be measured in term of plant biomass. We originally had included this statement to show the support of our findings to those revealed by Reider et al. 2018 Landsc. Ecol. We therefore have removed “(in terms of plant biomass)” from the discussion section. We have now edited the manuscript at line 251 to clarify this point and say: “..., and supports recent findings showing that species-area relationships are steeper (i.e. more extinction driven) in forested landscapes with a low-quality matrix, and shallower (i.e. less extinction driven) in those forested landscapes with a higher quality matrix⁵⁷.”

Comment:

“Line 276. ‘helps confirms’ -> ‘confirms’”

Response:

We thank the reviewer for this correction. We have fixed the typological error.

Comment:

“Lines 277-282. It would be nice to have a few lines here explaining how this work is an improvement on previous research. This paragraph gives the impression that most of it had been done before.”

Response:

We thank the reviewer for this suggestion. We have now amended our manuscript at line 285 to state: “Given that species are not homogeneously distributed throughout their ranges^{49,63}, this result specifically suggest that the condition of the matrix surrounding patches of suitable habitat is strongly correlated with extinction risk transitions in terrestrial mammals.”

Comment:

“Line 302. Suitable habitat was more important for what?”

Response:

We thank Reviewer 1 for this question. We have now amended our manuscript at line 305 to state: “..., but our results showed that the degree of fragmentation of suitable habitat, the extent of high human footprint values within the matrix, and the degree of isolation between patches of suitable habitat were more important predictors of extinction risk transitions than the change in high human footprint values (as represented by increases in high human pressure levels over time) within suitable habitat and the proportion of suitable habitat.”

Comment:

“Lines 319-321. This statement is too vague and doesn’t really say anything. Either elaborate on how exactly this can inform conservation, or just remove this line.”

Response:

We thank the reviewer for pointing this out. We have removed this statement from the manuscript as suggested.

Comment:

“METHOD (this is the part referred to in my main major concern comment)

Line 354. This is too vague. You need to explain fully how was the ‘expert information’ used and present it in an appendix and as part of the data made available.”

Response:

The source of this information, per Rondinini et al. 2011 Philos. Trans. R. Soc. B Biol. Sci., has not been fully added to our manuscript due to consolidation of data description with reference to the original source. However, we have now added detail to the methods starting from line 380 to state: “Textual descriptions of other habitat affinities for each species, derived from the

input of thousands of mammal experts belonging to more than 30 specialist groups of the IUCN Species Survival Commission (IUCN/SSC)⁵⁹, were also extracted from the IUCN database and input as quantitative data into the habitat suitability models.”

As noted in previous responses, and in the data availability section of our manuscript, the habitat suitability models were derived from a different study (Rondinini et al. 2011 Philos. Trans. R. Soc. B Biol. Sci.) and are only available upon request from the model developers. However, Dr. Carlo Rondinini, who led the development of the models and is co-author of our paper, agreed to share a small subset of these models for reviewers to assess. The subset of the habitat suitability models is available for download at https://drive.google.com/drive/folders/1bevNIINXr3WAF_hv1YIfD2944E7POrk0?usp=sharing. The code used for our spatial analyses is available at https://github.com/juanxramirez/Matrix_condition.

Comment:

“Lines 355 – 358. Explain how the habitat suitability levels were decided and defined.”

Response:

We thank Reviewer 1 for pointing this out. We have extended the methods section to explain how habitat suitability levels were decided and defined, while referring the reader to the original paper that developed and presented the habitat suitability models used in our study. We have now amended our manuscript in the methods section, starting from line 371, to read: “We used habitat suitability models developed by Rondinini et al.⁴⁹ to represent the extent of suitable habitat patches and the extent of the matrix of 4,426 out of 5,709 extant terrestrial mammals, corresponding to ~78% of all species in the group⁷⁸. The models were limited to occur within the known geographic range size of each species (the current “limits of distribution of a species, accounting for all known, inferred or projected sites of occurrence”, as defined by the IUCN Red List of Threatened Species^{78,79}), and built for the year 2000 at a spatial resolution of 300 m based on species’ elevation range and other habitat affinities, including preferred land cover types, tolerance to human impact, and relationship to water bodies. Species’ elevation range was incorporated into the habitat suitability models when known and recorded in the IUCN Red List. Textual descriptions of other habitat affinities for each species, derived from the input of thousands of mammal experts belonging to more than 30 specialist groups of the IUCN Species Survival Commission (IUCN/SSC)⁵⁹, were also extracted from the IUCN database and input as quantitative data into the habitat suitability models. The models ranked areas with three levels of habitat suitability: (i) high, representing primary habitat or preferred habitat where the species can persist; (ii) medium, representing secondary habitat where the species can occur but not persist without nearby high suitable habitat; and (iii) ‘unsuitable’, representing locations where the species is expected to occasionally or never be found. A subset of the models and their associated levels of habitat suitability were validated against available points of known species occurrences. Full details on the development of the models are available elsewhere⁴⁹.”

Comment:

“Line 362. This is not clear. What do you mean by ‘largest neighbour’? Aren’t all the pixels the

same size?”

Response:

We thank the reviewer for pointing this out. We were making reference to an 8-neighbourhood rule between the levels of habitat suitability, which was used when defining the continuity of each level of habitat suitability (using the 300 m resolution of the habitat suitability models). We have now amended the methods section of our manuscript at line 391 to read: “When delineating the levels of habitat suitability for each species, small contiguous groups of pixels (< 4 adjacent pixels of the same level of habitat suitability) were removed and replaced with the pixel value of the largest and nearest group of pixels, based on eight neighboring pixels of the same class.”

Comment:

“Line 382-383. Shouldn’t it be the other way around? i.e. large values of Euclidean distance representing LARGE rather than low degrees of isolation. Also, why not using the Euclidean distance between patches?”

Response:

We agree and thank the reviewer for pointing this out. Large values of the Euclidean distance represented large rather than low degrees of patch isolation, and we used the Euclidean distance between patches of suitable habitat through the surrounding matrix to account for patch isolation. We have now updated the text to reflect the suggested modification at line 411 to state: “Additionally, we calculated the average Euclidean distance between patches of suitable habitat through the surrounding matrix (i.e., the average Euclidean distance of all the pixels of unsuitable habitat from the nearest edge) to account for patch isolation (after⁴⁶). Here, large values of the average Euclidean distance represented high degrees of patch isolation, and small values represented low degrees of patch isolation.”

Comment:

“Lines 410-413. This sentence is not clear. I couldn’t understand what you meant by this. Please rephrase.”

Response:

Agreed. We thank the reviewer for highlighting this awkward wording. We have now rewritten this sentence starting at line 442 to read: “Based on previous studies^{43,83}, we used the extent of high human footprint values within the matrix as the extent of high pressure levels within species’ ranges has shown to be more sensitive to predict extinction risk than using mean values of human pressure within species’ ranges.”

Comment:

“Line 449. You can get rid of the first sentence and start with ‘Predictors included...’”

Response:

We thank the reviewer for pointing this out. We have removed this sentence as suggested.

Comment:

“Line 456. ‘avoid potential circularity’ is vague. Please explain this. Why is there circularity and how is this a way of avoiding it.”

Response:

We thank Reviewer 1 for pointing this out. We have now amended our manuscript in the method section, starting from line 502, to say: “Because the levels of habitat suitability are limited by the size of species’ geographic ranges, we did not include species’ range size as a predictor in order to avoid potential circularity in the estimation of extinction risk³⁸.”

REVIEWER 2 COMMENTS

Comment:

“The importance of the quality of the matrix of land-use types surrounding patches of focal habitat is being increasingly recognised. Many studies have found that the type of matrix can interact with and influence the effects of habitat loss and fragmentation. This manuscript investigates how the condition of the matrix, with regard to human pressures, effects the extinction risk of mammal species. By conducting this analysis on the majority of terrestrial mammals this study provides a global analysis which is useful in complementing the information gained from existing more localised studies. The study provides an important contribution to the building evidence on the significance of matrix quality when analysing habitat fragmentation. There are however a few issues that need to be addressed.

It is not completely clear how the range of a species is being defined. Does this use published distribution maps or is it based on the habitat suitability models?

As matrix quality is assigned based on the human footprint of unsuitable habitat within the range of a species, how the range is defined is important. The amount and proximity of the matrix habitat to suitable habitat will be influenced by how the range maps are created. For example, if the maps are created to a high resolution with habitat suitability accounted for this will be very different to polygons constructed linking occurrence records. Please can this be clarified or made more prominent if already included”

Response:

We thank Reviewer 2 for taking the time to improve our paper with such valuable feedback and for pointing this out. Starting from line 373, we have now fixed the text in the manuscript in a more nuanced way to clarify this point and say: “The models were limited to occur within the known geographic range of each species (i.e. the current “limits of distribution of a species, accounting for all known, inferred or projected sites of occurrence”, as defined by the IUCN Red List of Threatened Species⁷⁹), ...”

Comment:

“In the study the number of species with high-quality matrix and low-quality matrix were not equal but how were species with the two matrix types distributed with regard to geography and suitable habitat type (e.g. forest, grassland). If there are correlations between matrix quality and other biogeographic variables how does this influence the interpretation of the results?”

Response:

We thank Reviewer 2 for this very valid point. We have now produced two new figures (located in the supplementary information section) to show the distribution of the matrix condition in low-risk and high-risk species both globally and at the scale of individual biogeographic realms. We have also added a paragraph, starting from line 317, to discuss how such distribution is shown among biogeographic realms, stating that: “Although biogeography was not a key parameter for determining extinction risk transitions in our models, we found some differences in the way the matrix condition was distributed in low-risk and high-risk species among biogeographic realms. The Indo-Malay realm represented a particular case, with a highly right-skewed distribution (i.e. towards a higher extend of high human footprint values within the matrix) in species classified as low-risk, very similar to that shown in high-risk species. With ~87% of terrestrial mammals showing a low-quality matrix in the Indo-Malayan realm, this might indicate that species living in the Indo-Malayan realm are relatively more resilient to those human activities included in the human footprint, but more vulnerable to other threats (such as overexploitation, relevant in Southeast Asia⁷⁴), as also suggested by others⁴³. The Australasia realm also represented a particular case, with an approximately bimodal distribution of the matrix condition in low-risk species and a less right-skewed distribution in those classified as high-risk relative to that shown in other realms. Interestingly, the Australasia realm showed a lower proportion of species with a low-quality matrix (37.7%) when compared to other realms, suggesting that species restricted to this realm have relatively lower levels of human activity within their matrix. This is perhaps unsurprising given the fact that most recent declines of Australian terrestrial mammals have occurred in areas with low human population pressures, where native vegetation has not been significantly removed, particularly in the interior deserts and tropical savannas⁶⁶. In fact, the decline of most Australian species has been directly related to predation by introduced species (such as the feral cat, *Felis catus*, and the red fox, *Vulpes vulpes*) and changes in fire regimes^{66,67}, which are not represented in the human footprint maps used in our analyses.”

Although habitat suitability models were defined based on habitat affinities, these models do not provide insights on the type of suitable habitat of each species by themselves. It was therefore beyond the scope of this study to explore the distribution of the matrix quality with regard to suitable habitat type.

Comment:

“Was structural contrast between the suitable habitat and matrix considered? More contrasting suitable and unsuitable habitat types are expected to result in stronger fragmentation effects. To a certain extent the quality of the matrix from the perspective of the focal species would therefore depend on how different it is from the suitable habitat. The majority of this is likely captured by the human footprint data used but there may be differences between biomes.”

Response:

We thank the reviewer for this question. The structural contrast between patches of suitable habitat and the matrix was considered based on the levels of habitat suitability of each species. The levels of habitat suitability, as noted from line 383, are categorical and divided into: (i) high habitat suitability (i.e. primary habitat or preferred habitat where the species can persist),

(ii) medium habitat suitability (i.e. secondary habitat where the species can occur but not persist without nearby high suitable habitat), and (iii) unsuitable habitat (i.e. locations where the species is expected to occasionally or never be found). As mentioned in the methods section (lines 398-400), the extent of suitable habitat patches for each species was represented by combining high and medium habitat suitability, while the extent of the matrix by the level of unsuitable habitat.

In order to compare the predictive importance of habitat fragmentation (i.e. the degree of fragmentation and the degree of patch isolation) for the prediction of extinction risk transitions between species with a low-quality matrix (i.e. with a high-contrast matrix) and those with a high-quality matrix (i.e. with a low-contrast matrix), we have now redefined cutoff values for each of the levels of quality of the matrix (i.e. low-quality matrices and high-quality matrices). The cutoff values have now been derived from the positive and negative effect that the matrix condition had on the probability of high-risk transitions (see Fig. 3c). Starting from line 181, we have now amended our manuscript in the results section to state: "...we defined cutoff values for each of the levels of quality of the matrix based on the positive and negative effects that the matrix condition had on the probability of high-risk transitions (Fig. 3c). Low-quality matrices were therefore represented by species with extents > 84.2% of their matrix overlapping with high human footprint values ($n = 1,815$ low-risk species and 1,027 high-risk species), while high-quality matrices by those species with extents < 15.8% of their matrix overlapping with high human footprint values ($n = 60$ low-risk species and 29 high-risk species). We then built separate Random Forest models for each level of quality of the matrix in order to compare the relative importance of the degree of fragmentation of suitable habitat and the degree of isolation of patches of suitable habitat between species with a matrix of low-quality habitat and species with a matrix of high-quality habitat."

We agree with the reviewer that there may be differences between biomes with regard to the structural contrast between the suitable habitat and the matrix. Although it would have been interesting to explore this aspect, we believe it was outside of the scope of this study.

Comment:

"Please also see the comments below:

Line 78 – what are the indications from smaller scale studies?"

Response:

We thank Reviewer 2 for pointing this out. The indications for smaller scale studies were included in the discussion section of our manuscript. Starting from line 249, we state: "These findings are in line with previous studies showing that the use of the matrix is among the main determinants of the vulnerability of mammalian populations to local extinction in fragmented landscapes (e.g.^{25,55,56})."'

Comment:

"Line 361 – what size are the pixels?"

Response:

We thank the reviewer for pointing this out. The pixel size or spatial resolution of the habitat suitability models is 300 m, as mentioned at line 376.

Reviewers' Comments:

Reviewer #1:

Remarks to the Author:

The authors have satisfactorily addressed my concerns. I however wasn't able to run parts of their code because it is dependent on a python library that is only available to paying ArcGIS users and I don't hold a license for that. I would encourage users to use free, open source libraries.

I liked the naming of the variables tal and pascual :).

Reviewer #2:

Remarks to the Author:

The manuscript has been much improved by the revisions. The clarity has increased and the additions to the discussion and supplementary materials have provided the reader with more information, particularly around differences between regions.

Please find below some suggestions for small changes to the revised text -

Figure 3 - Please can the figure legend be expanded to explain how degree patch isolation should be interpreted (more or less isolated). This has already been done for fragmentation.

Lines 258 to 262 and Supplementary Figure 5 - Please can the details of this comparison be expanded, as it is currently difficult to follow the purpose of this test and the interpretation of the results.

Line 327 - Tropical forest is a structurally complex environment that contrasts highly with a cattle pasture or cropland matrix. I agree that comparison of matrix contrast between biomes is outside the scope of this study. However, it may be worth commenting on here as part of this discussion. There is a growing body of research examining how matrix type and contrast with the focal habitat can influence the strength of habitat loss and fragmentation effects at the local scale for a range of taxa.

Line 368 - I am not sure "by feral animals" is needed as predation risk in general may be elevated in the matrix.

Table 1 - the term "core" is used multiple times, but it is not clear where this is defined.

RESPONSE TO REVIEWERS

REVIEWER 1 COMMENTS

Comment:

“The authors have satisfactorily addressed my concerns. I however wasn't able to run parts of their code because it is dependent on a python library that is only available to paying ArcGIS users and I don't hold a license for that. I would encourage users to use free, open source libraries.

I liked the naming of the variables tal and pascual :).”

Response:

We thank Reviewer 1 for the excellent and detailed review. The manuscript has been immensely improved from the input provided, particularly around reproducibility.

REVIEWER 2 COMMENTS

Comment:

“The manuscript has been much improved by the revisions. The clarity has increased and the additions to the discussion and supplementary materials have provided the reader with more information, particularly around differences between regions.”

We thank Reviewer 1 for the excellent and detailed review. The manuscript has been improved considerably from the input provided, particularly around differences between biogeographic realms.

Comment:

*“Please find below some suggestions for small changes to the revised text -
Figure 3 - Please can the figure legend be expanded to explain how degree patch isolation should be interpreted (more or less isolated). This has already been done for fragmentation.”*

Response:

We thank Reviewer 2 for pointing this out. The figure legend of Figure 3 has now been expanded at line 836 to state: “High values of the degree of patch isolation represent high degrees of isolation between patches of suitable habitat.”

Comment:

“Lines 258 to 262 and Supplementary Figure 5 – Please can the details of this comparison be expanded, as it is currently difficult to follow the purpose of this test and the interpretation of the results.”

Response:

We thank Reviewer 2 for raising this very valid point. We have now expanded the manuscript at line 188 to read: “This indicates a greater effect of both the degree of fragmentation and the degree of isolation between patches of suitable habitat on the risk of extinction of those terrestrial mammals with a matrix of low-quality habitat.”

Comment:

“Line 327 – Tropical forest is a structurally complex environment that contrasts highly with a cattle pasture or cropland matrix. I agree that comparison of matrix contrast between biomes is outside the scope of this study. However, it may be worth commenting on here as part of this discussion. There is a growing body of research examining how matrix type and contrast with the focal habitat can influence the strength of habitat loss and fragmentation effects at the local scale for a range of taxa.”

Response:

We thank Reviewer 2 for this insightful comment. We have now amended the discussion of our manuscript stating at line 222 to state: “This suggests that the magnitude of the effects of fragmentation depend on the structural similarity between suitable habitat patches and the matrix, as also suggested by a growing body of evidence across multiple taxa on a local scale^{31,58}.”

Comment:

“Line 368 – I am not sure “by feral animals” is needed as predation risk in general may be elevated in the matrix.”

Response:

We thank Reviewer 2 for pointing this out. We have removed this statement from the discussion as suggested.

Comment:

“Table 1 – the term “core” is used multiple times, but it is not clear where this is defined.”

Response:

We thank Reviewer 2 for pointing this out. We have now amended Table 1 to state: “Average of the Euclidean distance from the edge to the ‘core’ (i.e. the interior) of each patch of suitable habitat”, to describe the variable degree of habitat fragmentation, and “Average of the Euclidean distance between patches of suitable habitat from the edge to the ‘core’ (i.e. the interior) of each area of unsuitable habitat”, to describe the variable degree of patch isolation.